# Partitioning of Small Hydrophobic Molecules into Polydimethylsiloxane in Microfluidic Analytical Devices

**DOI:** 10.3390/mi13050713

**Published:** 2022-04-30

**Authors:** Patrícia M. Rodrigues, Miguel Xavier, Victor Calero, Lorenzo Pastrana, Catarina Gonçalves

**Affiliations:** 1International Iberian Nanotechnology Laboratory, Avenida Mestre José Veiga, 4715-330 Braga, Portugal; patricia.rodrigues@inl.int (P.M.R.); miguel.xavier@inl.int (M.X.); victor.calero@inl.int (V.C.); lorenzo.pastrana@inl.int (L.P.); 2University of Minho, Gualtar Campus, 4710-057 Braga, Portugal

**Keywords:** polydimethylsiloxane, lab-on-a-chip, microfluidics, analytical devices, PDMS modification

## Abstract

Polydimethylsiloxane (PDMS) is ubiquitously used in microfluidics. However, PDMS is porous and hydrophobic, potentially leading to small molecule partitioning. Although many studies addressed this issue and suggested surface/bulk modifications to overcome it, most were not quantitative, did not address which variables besides hydrophobicity governed molecule absorption, and no modification has been shown to completely obviate it. We evaluated qualitatively (confocal microscopy) and quantitatively (fluorescence spectroscopy) the effects of solute/solvent pairings, concentration, and residence time on molecule partitioning into PDMS. Additionally, we tested previously reported surface/bulk modifications, aiming to determine whether reduced PDMS hydrophobicity was stable and hindered molecule partitioning. Partitioning was more significant at lower concentrations, with the relative concentration of rhodamine-B at 20 µM remaining around 90% vs. 10% at 1 µM. Solute/solvent pairings were demonstrated to be determinant by the dramatically higher partitioning of Nile-red in a PBS-based solvent as opposed to ethanol. A paraffin coating slightly decreased the partitioning of Nile-red, and a sol–gel modification hindered the rhodamine-B diffusion into the PDMS bulk. However, there was no direct correlation between reduced surface hydrophobicity and molecule partitioning. This work highlighted the need for pre-assessing the absorption of test molecules into the microfluidic substrates and considering alternative materials for fabrication.

## 1. Introduction

The first generations of microfluidic devices were fabricated from silicon and glass [1] using microfabrication techniques, such as photolithography [2] and etching, [3] which were already well-established within the microelectronics arena [4]. However, the use of these materials has been widely replaced by others that allow rapid prototyping and ease of manufacture [3,5]. Towards the end of the 20th century, microfluidics underwent a major expansion and evolution with the introduction of soft lithography and the subsequent ubiquitous use of polydimethylsiloxane (PDMS) in many research applications [6]. PDMS has since become the most widespread polymer for the fabrication of microfluidic devices owing to its optical transparency, chemical inertness, flexibility, gas permeability, and others [7,8].

However, PDMS is intrinsically hydrophobic and highly porous, which can lead to the undesired absorption of small molecules [9], potentially affecting the outcome of analytical experiments.

Toepke and Beebe [10] reported the absorption of Nile red (MW ~318 Da) into PDMS channel walls, which increased with repeated exposure. They also demonstrated that the absorption of quinine by PDMS depended on pH, as does its reversible repartitioning back into solution. Added to pH, the ionic strength of the solution can also considerably affect the absorption of molecules by PDMS [11]. Another study by Regehr et al. [12] showed that estrogen (hydrophobic; MW ~272 Da) was significantly more absorbed by PDMS when compared to prolactin (hydrophilic; MW ~24 kDa). The hydrophilicity of prolactin, along with its higher molecular weight, contributed to the lower absorption. The octanol/water (*o*/*w*) partition coefficient of a molecule can also help predict its tendency to be absorbed by PDMS [10,13]. More recently, Grant et al. described how the absorption of drugs into PDMS can significantly affect the performance of Organ-on-a-Chip devices. Here, the authors showed that fluorescein isothiocyanate (FITC), selected for having a molecular weight, structure, and hydrophobicity similar to amodiaquine, which is an anti-malarial drug, was absorbed by PDMS walls and diffused into its matrix [14]. Meer et al. also showed extensive absorption of drug molecules, which was more prominent for bepridil and verapamil, and studied the time-dependence upon exposure to PDMS [15].

The adsorption of biopolymers onto the surface of PDMS was also reported [16,17]. Work by Ken et al. [18] showed that cationic proteins adsorbed to PDMS due to electrostatic interactions. The authors also indicated the establishment of hydrophobic interactions and hydrogen bonds between the adsorbed and free solute.

PDMS has been linked to another significant artifact that might introduce bias to certain microfluidic applications [19,20,21,22]. Even after curing, PDMS contains uncrosslinked oligomers that can freely diffuse within the polymer bulk and leach out when in contact with solutions, thus introducing contaminants [12]. Previous studies aimed to remove residual oligomers from PDMS using methods including Soxhlet extraction [23], sequential extractions in organic solvents [24], and extended exposure to high temperatures [25]. There were also several investigations focused on the development of chemical/physical methods or coatings to mitigate or prevent the partitioning of hydrophobic molecules into PDMS. These methods include oxygen plasma treatment [26], silanisation [27,28,29], synthesis of silica nanoparticles using sol–gel treatments [30,31], and modification with surfactants such as Pluronic [32,33], among others. For example, Meer et al. tested the effect of commercial lipophilic coatings to reduce the absorption of hydrophobic molecules by PDMS [15].

Despite all the research conducted to date, there is still no consensus on the mechanisms governing molecule partitioning into PDMS, and the relationship between the molecule and PDMS properties is not yet fully understood. Here, we provide a further investigation of the factors affecting the partitioning of small molecules (Nile red, rhodamine B, and fluorescein) into the PDMS matrix, including solute/solvent pairings, solute concentration, and residence time. We show that it is important to consider solute/solvent pairings when assessing the absorption of hydrophobic molecules into PDMS. Our results also suggest that relative concentration changes are less prominent for increased concentrations of the tested molecules due to saturation of the PDMS matrix.

In addition, we tested different modifications to the PDMS surface and bulk, previously reported in the literature, that sought to reduce molecule partitioning. The stability of the modifications was assessed by monitoring the water contact angle (CA) for up to 21 days. The capacity to reduce molecule partitioning into PDMS was evaluated by flowing Nile red or rhodamine B through a serpentine microfluidic channel and measuring the fluorescence intensity by laser scanning confocal microscopy and fluorescence spectroscopy. We demonstrate that molecule partitioning into PDMS is not related only to the hydrophobicity of the surface, but it must also be associated with other properties, such as the PDMS porosity.

## 2. Materials and Methods

### 2.1. Materials

Sylgard™ 184 polydimethylsiloxane (PDMS) was purchased from Dow Corning (Midland, MI, USA). Nile red, rhodamine B, fluorescein sodium salt, phosphate-buffered saline (PBS), ethanol, acetone, Pluronic^®^ F127, paraffin wax, methylamine, (3-aminopropyl)triethoxysilane (APTES), and diisopropylamine, tetraethyl orthosilicate (TEOS) were purchased from Sigma-Aldrich (St. Louis, MO, USA). Ethyl acetate was obtained from PubChem (Bethesda, MD, USA).

### 2.2. Device Fabrication

Device master moulds were designed using CleWin4 from WieWeb Software (Hengelo, The Netherlands) and fabricated in polymethyl methacrylate using a FlexiCAM Viper computer numerical control micromilling machine (Eibelstadt, Germany) and tools from vhf camfacture AG (Ammerbuch, Germany). Devices were fabricated in PDMS by mixing the elastomer with a curing agent (10:1, *w*/*w*), degassed, poured over moulds, and cured at 65 °C for 12 h. The fabrication process varied slightly for some of the PDMS modifications described below. Inlets/outlets were opened using a 1.5 mm biopsy punch, and the PDMS slabs bonded to microscope glass slides using a PDC-002 Harrick Plasma Cleaner (Harrick Plasma, Ithaca, NY, USA).

### 2.3. Device Operations and Experimental Setup

The devices used for molecule partitioning studies consisted of a 35 cm long microfluidic serpentine channel with a 1 mm (width) by 0.5 mm (height) cross-section. Syringes containing the test solutions were connected to the devices using 1.6 mm outer diameter PTFE tubing, and flow was driven using a New Era NE-2000 syringe pump (New Era Pump Systems, Inc., Farmingdale, NY, USA). The outflow from the outlet was collected into an empty syringe to avoid solvent evaporation, and fluorescence was quantified by fluorescence spectroscopy to determine the ratio of fluorescence intensity before and after flowing through the device. In addition, the partitioning of small molecules into the channel walls was imaged and evaluated directly on-chip by laser scanning confocal microscopy.

### 2.4. PDMS Surface Modifications

#### 2.4.1. Pluronic^®^ F127

After fabrication, bonded PDMS devices were immersed in a 3% (*w*/*v*) Pluronic^®^ F127 solution in 1× PBS for 24 h. In order to evaluate the effect of surface oxidation by oxygen plasma on the modification with Pluronic^®^ F127, the same process was tested in PDMS devices straight after oxygen plasma activation [33].

#### 2.4.2. 3-Aminipropyltriethoxysilane (APTES)

For surface treatment with APTES, PDMS devices were cured at 80 °C for 1 h (instead of 65 °C for 12 h). The devices were then treated with oxygen plasma for 30 s and bonded. Immediately, the devices with an oxidised surface were immersed in a freshly prepared 1 or 10% (*w*/*v*) APTES solution in water/ethanol (50%, *v*/*v*) for 1 h to promote the formation of a covalent bond between the APTES molecules and the PDMS surface. The devices were finally washed with deionised water and dried under a stream of nitrogen [34,35].

#### 2.4.3. Paraffin

The fabricated devices were filled with melted paraffin wax (120 °C) for 1 min. The wax was then removed, and the device was allowed to cool down to room temperature [36].

### 2.5. PDMS Bulk Modifications

#### 2.5.1. Gradient-Induced Migration of Embedded Poloxamers

Prior to fabrication, a 200 mg∙mL^−1^ Pluronic^®^ F-127 solution in ethanol was added to the uncured PDMS at a 0.2% ratio (*v*/*w*), mixed thoroughly, and cured for 1 h at 75 °C. The PDMS devices were then oxygen plasma treated for 30 s and bonded to a glass slide, incubated in a dry oven at 75 °C for a further 30 min, and immersed in deionised water for 24 h [32].

#### 2.5.2. Thermal Aging

PDMS devices were placed in a dry oven at 100 °C for 7 or 14 days to promote the volatilisation and removal of low molecular weight PDMS chains from the polymer matrix. Following thermal aging, the devices were oxidised by oxygen plasma for 30 s and bonded to a glass slide [25].

#### 2.5.3. Solvent Extraction of Uncrosslinked Oligomers

Uncrosslinked PDMS oligomers were extracted from the elastomer bulk by immersing the cured elastomer in an extremely soluble solvent followed by cycles of decreasing solubility [24]. Devices were immersed in 150 mL diisopropylamine for 1 day, transferred to 150 mL ethyl acetate for another day, and finally to 300 mL acetone for 2 days (with acetone renewed after 24 h). The samples were dried in an oven at 100 °C for 2 days and oxidised using oxygen plasma for 30 s, and bonded.

#### 2.5.4. Sol–Gel

For sol–gel treatments, PDMS devices were cured at 80 °C for 1 h (instead of 65 °C for 12 h) and immersed in pure TEOS in a glass container for 60 min. The samples were constantly stirred for the first 5 min to avoid attaching to the glass surface. The samples were then washed with pure ethanol to prevent the formation of silica crystals, rinsed with deionised water, and immersed in a 4% (*v*/*v*) methylamine/water solution for 15 h. Methylamine and other contaminants introduced by the sol–gel treatment were eliminated from the bulk by further immersion in deionised water for 24 h (with the water changed 2×), which was shown to improve the biocompatibility of the elastomer [37]. Finally, the samples were dried at 80 °C for 1 h, oxidised by oxygen plasma for 30 s, and bonded.

### 2.6. Contact Angle Measurements

The water CA was measured using a Krüss Drop Shape Analyser-DSA 100 (Krüss GmbH, Hamburg, Germany) following the sessile drop method. An amount of 5 µL deionised water drops were dispensed from a motor-driven syringe onto the device surface, imaged using an integrated camera, and the water CA was determined using DSA3 software.

### 2.7. Fluorescence Spectroscopy

The retention of small molecules by PDMS was estimated by determining the fluorescence intensity ratio of the solutions before and after flowing through the PDMS device. The fluorescence intensity of Nile red in ethanol (λex: 543 nm; λem: 627 nm), Nile red in a 1:3 ethanol/PBS solution (λex: 584 nm; λem: 650 nm); fluorescein in PBS (λex: 485 nm; λem: 515 nm), and rhodamine B in PBS (λex: 550 nm; λem: 580 nm), was measured using a BioTeK^®^ Synergy H1 (Winnoski, VT, USA) microplate reader. Samples were measured in triplicate using black Greiner Bio-One 96-well plates. The fluorescence excitation and emission maxima of each solution were determined using a FluoroMax-4 spectrofluorometer from Horiba (Kyoto, Japan) (Appendix A).

### 2.8. Laser Scanning Confocal Microscopy

The partitioning of fluorescent molecules into the PDMS matrix was also visualised directly within the device by laser scanning microscopy images taken at multiple focal planes defined within the limits of each channel wall. A Zeiss laser scanning confocal microscope LSM780 (Oberkochen, Germany) equipped with an EC Plan-Neofluar 10x objective (N.A. 0.3) was used. The 488 nm and 513 nm argon lasers were used to image fluorescein and rhodamine B, respectively, while Nile red was imaged at 651 nm using a diode-pumped solid-state laser. The images were analysed using Zen 2010 software from Zeiss and ImageJ.

### 2.9. Statistical Analysis

The results are displayed as Mean ± SD, unless otherwise stated, using GraphPad Prism 9 (San Diego, CA, USA). Data distributions were tested for normality using the Shapiro–Wilk test. The statistical significance was tested using a Student′s t-test or a one-way analysis of variance (ANOVA) with Tukey′s post hoc test for samples that followed a normal distribution and a Kruskal–Wallis test followed by Dunn′s multiple comparisons for samples that did not follow a normal distribution.

## 3. Results and Discussion

Molecule partitioning into PDMS can be used to the benefit of the user [9]. However, it might also lead to misleading experimental outcomes when it results in unpredictable changes to the concentration of test solutions. To understand the dependence of this phenomenon on different factors, such as solute/solvent pairings, solute concentration and residence time, a series of experiments were conducted aiming to determine the conditions where molecule partitioning into PDMS can be more significant or even problematic.

### 3.1. Molecule Partitioning as a Function of Solute/Solvent Pairings

The partition coefficient of a compound can be used as an indicator of its hydrophilicity. It is defined as the ratio at thermodynamic equilibrium between the concentration of a compound in a hydrophobic solvent (with octanol being chosen as the standard nonpolar solvent due to its amphipathic structure) and its concentration in a hydrophilic solvent (water) [38]. In practice, it is commonly used in its logarithmic form and referred to as the octanol-water partition coefficient ($K_{OW}$), with high $K_{OW}$ indicating high hydrophobicity.

In order to assess how a molecule $K_{OW}$ influenced the partitioning of small fluorescent molecules into PDMS, rhodamine B (MW=479.02 Da) and fluorescein (MW=376.27 Da) solutions in 1× PBS were used given their similar molecular weight and different $K_{OW}\;$at 2.74 and −0.67, respectively [13].

Confocal microscopy images of the PDMS channels showed that the partitioning of molecules into PDMS increased with the molecule′s $K_{OW}$. Figure 1a shows that the channel walls became fluorescent when rhodamine B flowed as opposed to fluorescein. As previously reported [9], $K_{OW}$ greater than 1 indicates a tendency of partitioning and diffusing into PDMS, which constitutes a hydrophobic phase. This corroborated the reduced concentration of rhodamine B detected at the outlet, quantified by spectrophotometry (Figure 1b). Interestingly, a study by Wang et al. [13] indicated that all compounds with a $K_{OW}$ greater than 2.67 would be highly absorbed by PDMS. However, a later study by Meer et al. [15] was unable to find a clear correlation between molecule $K_{OW}$ and absorption, suggesting that other parameters such as topological polar surface could be involved.

In order to determine the effect of solute/solvent pairings on the partitioning of small hydrophobic molecules into PDMS, the Nile red ($K_{OW}=5.0$) was flowed diluted in either ethanol or an ethanol/PBS solution (1:3 *v*/*v*). Nile red is hydrophobic and thus stable in ethanol [39], which contains polar (OH) and nonpolar (C_2_H_5_) groups. Importantly, PBS as a solvent represents a more physiologically relevant scenario and is more likely to be used in biological studies than ethanol, hence the 25% ethanol in the PBS mixture used in this study. When Nile red in ethanol/PBS flowed through the microchannel, there was a dramatic increase in the molecule partitioning into PDMS, and only a negligible fraction of Nile red remained in the solution (Figure 1b). Crucially, in these conditions, the partitioning of Nile red was so significant that it occurred immediately after the first instants of contact with PDMS (Figure 1a). These results highlighted the importance of considering solute/solvent pairings in studies using PDMS-based devices and are in agreement with previous studies [10]. As a practical example, Nile red is a fluorescent marker commonly used to stain oil in nanoemulsions targeting the delivery of hydrophobic molecules via oral administration [40]. During in vitro digestion studies, Nile red incorporated in oil droplets would be typically released and integrated into mixed micelles that become dispersed in water-based simulant fluids. In a PDMS-based digestion simulator, the partitioning of Nile red into PDMS would most likely alter result interpretation.

### 3.2. Molecule Partitioning as a Function of Residence Time

In microfluidics, small channel dimensions and high throughput typically imply short residence times inside devices. However, certain applications require long incubation times and thus lower flow rates and/or larger channels. Here, 1 µM Nile red in ethanol/PBS (1:3 ratio) was flowed at 400 or 1000 µL∙h^−1^, corresponding to a 2.5-fold difference in residence time. The amount of Nile red that remained in the solution was significantly higher at the highest flow rate (Figure 2; 73.7% vs. 28.6%, p<0.01). This shows that it is particularly important to monitor the partitioning of small molecules in PDMS-based devices when long residence times are anticipated.

### 3.3. Molecule Partitioning as a Function of Concentration

It is intuitive that higher solute concentrations should lead to higher absorption rates by PDMS due to the formation of a greater concentration gradient. Figure 3a shows that when 20 µM rhodamine B in PBS flowed through the device, there was a significantly higher molecule partitioning when compared to the same solution at 1 µM. Importantly though, in relative terms, the percentage of solute that remained in the solution was higher for the 20 µM solution, with a relative reduction of just 10%, contrasting with a reduction of around 90% observed for the 1 µM solution (Figure 3b).

These results indicate that although more solute is absorbed by PDMS at higher concentrations, in relative terms, at lower concentrations, molecule absorption by PDMS becomes more significant. This is critical given that at low concentrations, solutes are often close to the limits of detection of analytical methods. Thus, such reductions in concentration could lead to a complete failure to detect molecules of interest in analytical microfluidic devices. At later time points, the relative concentration of rhodamine B at the outlet increases, suggesting that as PDMS becomes saturated, the gradient decreases, leading to lower partitioning of molecules.

### 3.4. PDMS Modifications

#### 3.4.1. Contact Angle Measurements

Aiming to reduce the partitioning of small molecules into PDMS, a series of surface and bulk modifications reported in the literature were tested and compared. To monitor the stability of the modifications, the water CA of the PDMS surface was measured over time up to 21 days. PDMS is a hydrophobic elastomer with a water CA around 110° [41].

Figure 4 shows that all the modifications tested, except for the surface modifications with Pluronic^®^ F127 and paraffin, led to a sharp decrease in the water CA of PDMS, in some cases down to values as low as 10°. However, there was a general tendency for the water CA to return to values rounding those of unmodified PDMS (around 100°) over time, in agreement with previous research [42]. This reversal of PDMS to its original surface properties is usually attributed to the reorientation of polar groups from the surface into the bulk and to the migration of uncrosslinked PDMS oligomers from the bulk to the surface [24,25,32].

Interestingly, the removal of uncrosslinked PDMS oligomers through thermal aging and solvent extraction methods did not reduce the PDMS hydrophobic recovery after oxygen plasma treatment. This may suggest an incomplete extraction of uncrosslinked oligomers leading to their diffusion through the PDMS bulk to restore a thermodynamically stable surface [25]. In addition, our devices were stored at room temperature in contact with air. Controlled storing conditions of the PDMS devices could also help maintain the stability of the modifications over time, as pointed out in previous publications [42]. Nevertheless, the hydrophobicity renewal rate was more or less marked depending on the modification. For example, the surface treatment with Pluronic^®^ F127 and oxygen plasma showed a relatively stable modification in the long term, where the water CA remained under 50° during the first week, increasing to 75° after 3 weeks.

#### 3.4.2. Molecule Partitioning

The effect of the modifications aiming to mitigate or prevent the partitioning of small hydrophobic molecules into PDMS tested here was evaluated by flowing Nile red in ethanol/PBS (1:3) through the PDMS microchannels. Figure 5a shows the percentage (%) of Nile red that remained in the solution after flowing through unmodified PDMS microchannels and PDMS microchannels following the various modifications. The experiments were conducted on the same day that the modifications were finalised to ensure that the experimental results were not affected by the recovery of the PDMS surface to its original hydrophobicity. The analysis of Figure 4 and Figure 5a demonstrates that the decrease in the PDMS surface hydrophobicity achieved by some surface and bulk modifications did not translate into a reduction in Nile red partitioning. In fact, some modifications led to an increase in molecule partitioning (e.g., bulk modification with Pluronic^®^ F127, solvent extraction, and 14-day thermal aging). These results suggest that the partitioning of small hydrophobic molecules into PDMS is not dominated by the hydrophobic nature of the PDMS surface and might be more affected by the porosity of the elastomer and other properties such as topological polar surface area, as suggested in previous publications [15,43].

Three-dimensional Hansen solubility parameters (HSP) were also shown to offer good predictive power of the swelling of PDMS when in contact with water, ethanol, butanol and toluene [44]. HSP uses three solubility parameters for dispersion, polarity and hydrogen bonding and could potentially be used to predict the absorption of small molecules by PDMS with higher accuracy than an analysis based solely on molecule $K_{OW}$ and surface hydrophobicity. Although studies of HSP exceeded the scope of this work, we believe that future studies determining solubility ratings could aid in predicting the absorption of small molecules by PDMS and also which modifications could successfully reduce molecule absorption.

We tested a PDMS bulk modification with TEOS, aiming to fill the PDMS matrix with silica particles. Although this modification did not lead to a reduction in Nile red or rhodamine B partitioning, it significantly impaired the diffusion of rhodamine B into the polymer matrix, which is in agreement with previous work [37]. A comparison between Figure 5b,c shows that after the sol–gel treatment, the molecules concentrated at the surface of the PDMS, in opposition to unmodified PDMS, where the molecules were able to migrate through the PDMS bulk.

In addition, although the paraffin modification led to a slight increase in the water CA, it was the only modification that provided a reduction (though not statistically significant *p* = 0.07) of the partitioning of Nile red into PDMS. These results are in agreement with the experiments detailed above, supporting our hypothesis that the partitioning of small hydrophobic molecules is not directly correlated to the hydrophobicity of the PDMS surface and that other factors play a significant role in the absorption mechanism.

Importantly, when the device was visualised by confocal microscopy, negligible fluorescence was found in the PDMS channel walls (Appendix A), although only around 12% of Nile red remained in the solution collected at the outlet. We hypothesise that paraffin might cause fluorescence quenching of rhodamine B molecules, as previously reported for green fluorescent protein (GFP) [45]. Hence, we emphasise the importance of characterising molecule partitioning by using fluorescence techniques that can quantify the molecule concentration before and after solutions flow through PDMS devices.

## 4. Conclusions

This work reiterated the capacity of PDMS to work as an efficient solid solvent. While this may be used to the benefit of the user, it is more often detrimental, particularly in applications where precise quantification of an analyte or low concentrations are required. When the partitioning of molecules of interest may pose a problem, a case-by-case analysis should be carefully followed, as it depends on a variety of factors, including solute/solvent pairings, concentration, and residence time, as demonstrated in the present work.

Critically, molecules with high octanol/water partition coefficients showed a significant partitioning into PDMS to an extent at which their remaining concentration in solution became negligible. However, the use of higher flow rates can reduce this effect. It is at lower analyte concentrations that this can be more significant as the analyte is extensively absorbed before the concentration gradient reduces sufficiently to slow down molecule partitioning.

The bulk and surface treatments tested here as modifications to PDMS did not significantly reduce the partitioning of the test molecule—Nile red. Our experimental results indicate that the hydrophobic nature of the PDMS surface does not dominate molecule partitioning. This suggests that it is important to consider other factors beyond wettability, such as the elastomer porosity or topological surface area, when studying or aiming to reduce molecule absorption in PDMS-based devices.

Future work should focus on the possibility of pre-saturating the PDMS matrix and also study the dependence of molecule absorption on material porosity and other factors. Hansen solubility parameters could also be explored, aiming to predict the solubility of certain molecules in PDMS beyond molecule $K_{OW}$ and surface hydrophilicity. Ultimately, this study indicates that when designing devices for analytical purposes and the partitioning of molecules of interest is deemed to be problematic, alternative materials to PDMS should be considered for fabrication.

## Figures and Tables

**Figure 1 micromachines-13-00713-f001:**
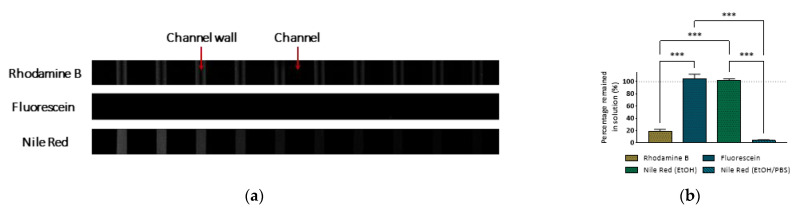
(**a**) Confocal microscopy images of PDMS channel walls following continuous flow of rhodamine B and fluorescein in 1× PBS and Nile red in ethanol/PBS (1:3 *v*/*v*). (**b**) Relative concentration ratio (expressed as a percentage) of 1 µM rhodamine B, 1 µM fluorescein, and 1 µM Nile red in ethanol or ethanol/PBS (1:3 ratio) that remained in solution following 90 min of continuous flow through the PDMS microchannels at 96 µL∙h^−1^. Values show Mean ± SD from at least three independent measurements (*** *p* < 0.001).

**Figure 2 micromachines-13-00713-f002:**
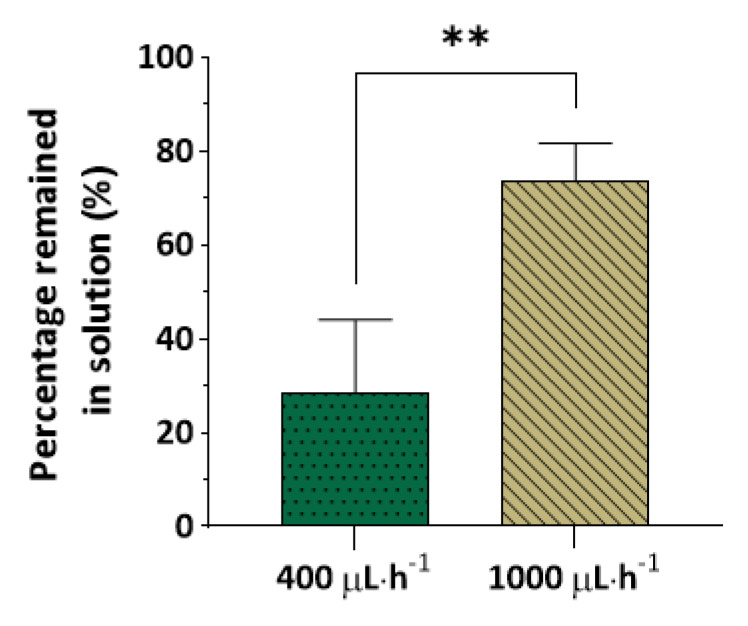
Relative concentration ratio (expressed as a percentage) of 1 µM Nile red in ethanol/PBS (1:3 *v*/*v*) that remained in solution following continuous flow through PDMS microchannels at 400 µL∙h^−1^ or 1000 µL∙h^−1^. Values show Mean ± SD from at least three independent measurements (** p<0.01).

**Figure 3 micromachines-13-00713-f003:**
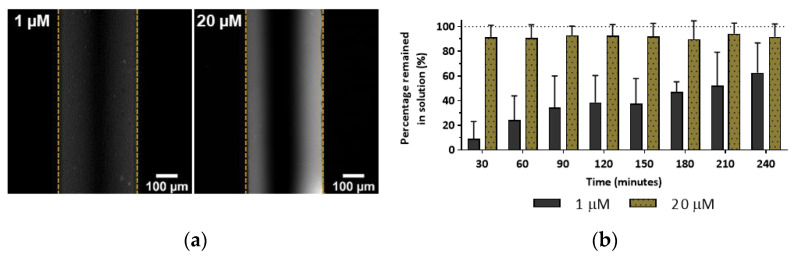
(**a**) Confocal microscopy images of PDMS channel walls following continuous flow of 1 µM or 20 µM rhodamine B solutions. (**b**) Relative concentration ratio over time (expressed as a percentage) of rhodamine B that remained in solution following continuous flow through the PDMS microchannels 400 µL∙h^−1^. Values show Mean ± SD from at least three independent measurements.

**Figure 4 micromachines-13-00713-f004:**
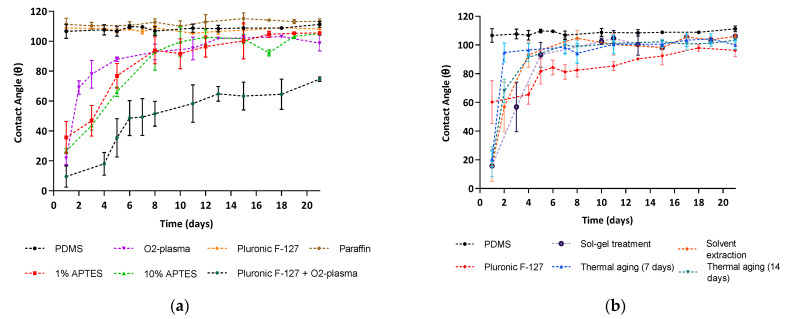
Water contact angle measurements as a function of time of unmodified PDMS and PDMS following (**a**) surface and (**b**) bulk modifications. Values show Mean ± SD from at least three independent measurements.

**Figure 5 micromachines-13-00713-f005:**
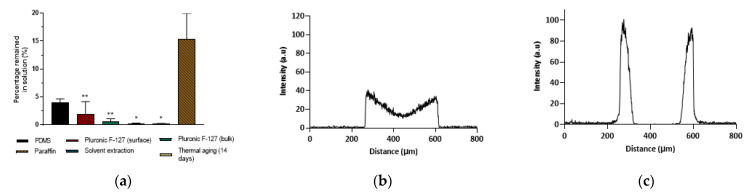
(**a**) Relative concentration ratio (expressed as a percentage) of Nile red that remained in solution following continuous flow through unmodified PDMS microchannels or PDMS microchannels modified with Pluronic^®^ F127 (surface and bulk modifications), paraffin, thermal aging, and solvent extraction. Values show Mean ± SD from a minimum of two independent experiments (** *p* < 0.01; * *p* < 0.05 vs. paraffin modification). (**b**,**c**) Fluorescence intensity profiles obtained from the confocal micrographs following continuous flow of a 20 µM rhodamine B solution through (**b**) unmodified PDMS microchannels and (**c**) bulk-modified PDMS with sol–gel treatment using tetraethoxysilane (TEOS).

## Data Availability

The complete dataset supporting this article is available from 10.5281/zenodo.5825722, accessed on 29 March 2022.

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
