# Peer review of "Partitioning of Small Hydrophobic Molecules into Polydimethylsiloxane in Microfluidic Analytical Devices"

_micromachines, 2022, doi:10.3390/mi13050713_

Round 1
Reviewer 1 Report
Dear authors,
Please consider the suggested comments to improve the quality of the current version of the manuscript:
- The abstract needs to be incorporated with the gist of the complete work in the manuscript. In the current version of the abstract, only the summary of the work is mentioned in a very broad view. Instead, please include the specific details of the research study presented in the manuscript.
- The abstract of a scientific paper should precisely mention the specific scientific question that is answered, and the scientific conclusions made based on the experimental results. Please explain and provide the specifications of the scientific questions answered in the proposed study.
- In the abstract, please include the details of the limitations of the previous research studies/technologies made in the proposed domain of study. Also, specify the scientific advancements made in the current/proposed study to overcome those limitations.
- In the current version of the abstract, the scientific conclusions made based on the experimental results attained were not clearly mentioned (in lines 23-26). Please include the details of conclusions driven from the experimental results.
- In the introduction, please include additional details of the advanced studies performed by peers on the ‘Partitioning of hydrophobic molecules’, to guide the reader to understand the importance of the study performed. Also, include corresponding references in the text when mentioning the details.
- In the introduction, please include the knowledge gaps existing between the current proposed study and prior studies performed in this field. Very importantly, please specify the need for the current work presented in the manuscript.
- In the last paragraph of the introduction, kindly include the details of the broader impacts on the study made and the results achieved. It is very important to provide the future scope of the study performed to make a strong impact on the readers of the research performed/Study proposed.
- Please update the manuscript with high-quality images especially for the plots as the images in the current version of the manuscript are difficult to study in detail.
- In section-3.4 (PDMS Modifications), the current version of the manuscript is only explaining and detailing the results (especially the plots for figures-4 &5). However, please provide scientific reasoning for the results obtained. Please use the ongoing studies published by peers with appropriate references to support your arguments and statements.
- Please include more appropriate references from the studies that are performed more recently. In the current version of the manuscript, there are some references that are not recent publications.
- Please use the proper format for writing and mentioning the mathematical equations. Kindly refer to the author's instructions to understand the formatting process during the submission of the manuscript. Please use the link below for author instructions: https://www.mdpi.com/journal/micromachines/instructions
- Please revise the manuscript with English grammar. There are a few places that the manuscript needs to be improved with respect to English writing.
Reviewer 2 Report
Motivation/Goal of the work:
PDMS is the most ubiquitous material used to fabricate microfluidic devices, this material has distinct advantages, as well as disadvantages – one particularly challenging disadvantage is the absorption of small molecules. PDMS is hydrophobic and highly porous, so it absorbs small molecules, which affect experimental outcomes, thus identifying ways to minimize absorption, would be ideal. This work evaluated different factors affecting small molecule absorption: solute/solvent pairings, solute concentration, and residence time. Part of the evaluation included testing the effect of PDMS surface treatment and PDMS bulk modifications on small molecule absorptions.
Summary of work
The authors did an evaluation of three small molecules, with molecular weights of the same order of magnitude, with differing octanol-water partition coefficients. These three small molecules (fluorescein, rhodamine B, and nile red) were assessed in connection with their absorption in PDMS, and how modifications of PDMS (either as a surface treatment or in bulk) would impact the wettability, contact angle of water, on PDMS and if these changes reduced the absorption of hydrophobic small molecules.
General evaluation of the work:
This work provides a protocol to measure small molecule absorption in PDMS with different small molecules and different PDMS surface treatment/bulk modifications. It is key to note that the authors assessed and compared the wettability, often a measurement of hydrophilicity, and corresponding absorption of the hydrophobic molecules across the different treatments. This element importantly highlights that these two measurements are not always correlated in outcomes, thus there is a need to assess absorption beyond wettability of a material. The consideration of residence time, and concentration effects importantly highlight the impact of the specific experimental conditions and how this can significantly impact the concentration change (both as a function of total amount absorbed vs. percent change). The paper clearly, and effectively conveyed the work that was done. Additionally, the methods used for PDMS modification were generally simple modifications to the material and would be easily adoptable by many labs were they successful at reducing the absorption of the small molecules into the material. Interestingly, none of the surface treatment/bulk modifications this paper evaluated can reduce small molecule absorption effectively, which implies the necessity of microfluidics fabricated with alternative materials and a prerequisite of assessing the absorption of molecules into the material used.
While this work evaluated an important topic, of the absorption of hydrophobic molecules in PDMS, it limited the discussion around mechanisms of why the absorption was occurring. The discussion around how surface treatments, even ones that promote a lower contact angle for increased time, had little to no impact on absorption could have been described in more detail. The only presentation of some level of mechanism was the octanol-water partition coefficient, though there was little information in regard to this. Some mechanistic evaluation, consideration, or discussion may be helpful in providing a better context of how to predict the solubility of different hydrophobic molecules. The authors use the Kow partition coefficient, which enables quantification of hydrophobicity, however, there may be other measures (for example Hansen Solubility parameters) that may be better to describe the absorption of small molecules into PDMS, as a “solid solvent”. While the authors highlight a critical point, which is the need to assess the absorption of small hydrophobic molecules into the choice material for microfluidic studies (especially if using PDMS), this point has been highlighted previously in other works that the authors reference directly.
Specific comments and change suggestions:
Line 102: is it supposed to be “… and fluorescence was quantified by fluorescence spectroscopy”?
There was little discussion of interaction between molecules and different materials/solvents, particularly elements to elucidating some prediction of the portions of molecules being absorbed. For example, work that provides a bit more evaluation or context of the molecule characteristics being absorbed into PDMS:
“van Meer, B. J., de Vries, H., Firth, K., van Weerd, J., Tertoolen, L., Karperien, H., Jonkheijm, P., Denning, C., IJzerman, A. P., & Mummery, C. L. (2017). Small molecule absorption by PDMS in the context of drug response bioassays. Biochemical and biophysical research communications, 482(2), 323–328.”
Because this work largely uses the bases of hydrophobicity of the molecules, but the contact angles should be quite low on the day of testing the molecular partitioning, the reasoning that it is purely the porosity of the material (Line 320) seems a bit limited.
It would be helpful to clarify the different device curing and treatments, in case future readers wanted to replicate or repeat the different methods of modification – for example in:
- Section 2.4 PDMS Surface modifications, it describes devices being cured for 1-hour at 80C (for APTES treatment, 2.4.2), which as a read is assumed that it would be after already curing and preparing/bonding the devices as stated in section 2.2. Is this accurate? If so, please update text to make it clear, that the devices were already generated and a subsequent 80C treatment was done.
- Similarly for 2.5 PDMS Bulk modifications, it sounds like the 2.5.1: Gradient-induced migration of embedded poloxamers was done to a PDMS sample and not a device specifically, though for thermal aging and solvent extraction, a device is specifically stated in the methods. Clarification, of 2.5.1. whether there was a device made, or if it was just a layer of PDMS, would be useful.
While the description of non-consistent contact angles is useful, this has previously been done in a variety of settings and should be mentioned or cited, and potentially put in context as the results seen by the authors in terms of “reversal” were not unexpected. Additionally, returning back to a baseline Contact Angle, is well known to occur with PDMS relatively quickly unless there is a storage condition that ensures the surface to retain the modification. One example: “B. Kim, E. T. K. Peterson and I. Papautsky, "Long-term stability of plasma oxidized PDMS surfaces," The 26th Annual International Conference of the IEEE Engineering in Medicine and Biology Society, 2004, pp. 5013-5016, doi: 10.1109/IEMBS.2004.1404385.” Due to this, it may also be beneficial for the authors to highlight the storage conditions of the samples that were tested at different time intervals for the contact angle measurements.
From the data one of the considerations is that the researchers could pre-saturate the PDMS with the molecule of interest (like what is being done in Figure 3a) to demonstrate this as a potential way to eliminate any further absorption. This may result in release of the specific molecule – but may be something that should potentially be considered, discussed in this manuscript, or they may just want to try in future work.
The only data or discussion that seemed to indicate some reduction of absorption aside from paraffin was the sol-gel treatment. While the reviewer does not think it is critical to repeat an experiment, it is unclear why there is not more of a discussion or evaluation of this treatment in figure 5a, to examine the percentage remaining in the solution.
